# Tear Cytokine Changes up to One Year After Allogeneic Hematopoietic Stem Cell Transplant: Effect of Daily Topical Cyclosporine-A 0.1% Emulsion

**DOI:** 10.3390/ijms26125915

**Published:** 2025-06-19

**Authors:** Louis Tong, Yu-Chi Liu, Sharon Wan Jie Yeo, Chang Liu, Isabelle Xin Yu Lee, Yeh Ching Linn, Aloysius Ho, Hein Than, Jeffrey Kim Siang Quek, William Ying Khee Hwang, Francesca Lorraine Wei Inng Lim, Li Lim

**Affiliations:** 1Corneal and External Eye Disease Service, Singapore National Eye Centre, 11 Third Hospital Avenue, Singapore 168751, Singapore; liu.yu.chi@snec.com.sg (Y.-C.L.); lim.li@singhealth.com.sg (L.L.); 2Ocular Surface Research Group, Singapore Eye Research Institute, 20 College Road Discovery Tower Level 6, The Academia, Singapore 169856, Singapore; sharon.yeo.w.j@singhealth.com.sg (S.W.J.Y.); liu.chang@seri.com.sg (C.L.); isabelle.lee.x.y@seri.com.sg (I.X.Y.L.); 3Eye-Academic Clinical Program, Duke-National University of Singapore Medical School, 8 College Road, Singapore 169857, Singapore; 4Department of Ophthalmology, Yong Loo Lin School of Medicine, National University of Singapore, 10 Medical Dr., Singapore 117597, Singapore; 5Department of Haematology, Singapore General Hospital, 31 Third Hospital Ave, Singapore 168753, Singapore; linn.yeh.ching@singhealth.com.sg (Y.C.L.); aloysius.ho.y.l@singhealth.com.sg (A.H.); hein.than@singhealth.com.sg (H.T.); jeffrey.quek.k.s@singhealth.com.sg (J.K.S.Q.); william.hwang.y.k@singhealth.com.sg (W.Y.K.H.); francesca.lim.w.i@singhealth.com.sg (F.L.W.I.L.)

**Keywords:** graft-versus-host disease, ocular surface disease, dry eye disease, tear disorders, cytokines, therapy, prophylaxis, immunology

## Abstract

Purpose: To profile tear cytokine changes in Allogeneic Hematopoietic Stem Cell Transplant (HSCT) patients after instillation of daily topical cyclosporine-A 0.1% cationic emulsion. Methods: Participants in a longitudinal study were given cyclosporine eyedrops daily from 3 to 5 weeks before and 3 months, 6 months, and 12 months post-HSCT. The outcomes included tear cytokine concentration assayed by the Proximity Extension Assay O-linked target 96 platform. The patients were divided into two groups: Group 1 (*n* = 8 conjunctival CD4 cells responding to cyclosporine) and Group 2 (*n* = 5 conjunctival CD4 cells not suppressed after cyclosporine, where patients were non-compliant with cyclosporine). All participants had a standardized clinical examination, including meibomian gland evaluation and tear breakup times. Results: The levels of 38 cytokines/chemokines showed significant changes (*p* < 0.05) over time, and in many, the elevation was marked at one year. These include gamma-interferon, CXCL9, CCL3, and CCL4 (all *p* < 0.0001). For gamma-interferon, there was significant interaction between group and time at 1 year (*p* = 0.022), where the cytokine was significantly suppressed in Group 1. Four other cytokines showed significant group and time interaction at 1 year: FGF23, FGF5, LIFR, and Enrage (all *p* < 0.05). All patients had either withdrawal or a reduction in systemic immunomodulation between 6 months and 1 year. We found several cytokines to be associated with changes in tear osmolarity or symptom scores. Conclusions: HSCT induces significant elevation of 38 tear cytokines/chemokines even without the occurrence of ocular graft-versus-host disease when systemic immunosuppression is reduced within the first year. Topical daily cyclosporine eyedrops can reduce some pro-inflammatory tear cytokines.

## 1. Introduction

Dry eye disease (DED) is a common ocular sequela of graft-versus-host disease (GVHD) after allogeneic hematopoietic stem cell transplantation (allo-HSCT), with one study showing more than 40% of allo-HSCT patients developing chronic ocular GVHD (according to both NIH consensus criteria and the International Chronic Ocular GVHD Consensus Group) [1,2,3]. Acute GVHD happens during the first 100 days following HSCT, but chronic GVHD occurs beyond that time. Approximately 7.5 months passed on average between allo-HSCT and the start of ocular GVHD [4,5]

The pathogenesis of DED in GVHD is characterized by a T cell-mediated infiltration and inflammation of the lacrimal gland, conjunctiva, and ocular surface. This process leads to fibrosis of the lacrimal gland and conjunctiva, a reduction in the number of conjunctival goblet cells, and a decrease in tear production [6,7,8]. GVHD also causes meibomian gland obstruction and anterior and posterior blepharitis [9,10]. Previously, we showed that people who underwent HSCT displayed one of two patterns of change in conjunctival CD4 T cells, depending on their compliance with cyclosporine [11].

It is advantageous to investigate preventive measures for DED in this population, given the high incidence of DED in allo-HSCT patients. It has been observed that topical CsA is both safe and effective in treating DED linked to chronic GVHD [12,13,14]. However, there are only a few reports on topical cyclosporine-A in the prophylaxis of ocular GVHD [15,16,17].

Since cytokines in the tear drive ocular surface inflammation, we are interested in determining whether concentrations of various cytokines change with time after HSCT and whether instillation of topical cyclosporine influences these concentrations.

In this prospective study, our primary aim was to profile tear cytokine concentrations in allogeneic HSCT patients up to one year and to investigate the effect of daily topical cyclosporine-A 0.1% cationic emulsion on tear cytokines. A secondary aim was to evaluate whether clinical symptoms and signs of dry eye in the participants would significantly change when levels of tear cytokines were altered during the follow-up.

## 2. Methods

### 2.1. Design

This is an open-label single-arm interventional study performed at the Singapore National Eye Centre (SNEC) in collaboration with the Hematology Department at the Singapore General Hospital (SGH).

### 2.2. Study Population

Patients who underwent allo-HSCT at the Hematology Department, SGH, between November 2019 and September 2021 were recruited. Every participant signed an informed consent form. The SingHealth Centralised Institutional Review Board prospectively examined and authorized our work, which aligned with the principles of the Declaration of Helsinki on human research (reference number: 2019/2635). The Clinical Trials database registration number for the trial is NCT04636918; the URL is https://clinicaltrials.gov/ct2/show/NCT04636918?cond=ocular+Graft+Versus+Host+Disease&cntry=SG&draw=2&rank=2 (accessed on 16 June 2025).

### 2.3. Eligibility

The participants in the trial had undergone HSCT and were 13 years old or older, capable of providing informed consent, and able to meet study evaluation conditions. The following patients were excluded from the study: age below 13 years, the presence of other concurrent ocular or systemic disease that may interfere with study results, non-resident patients who were unable to complete follow-up, and existing users of topical cyclosporine eyedrops. During the COVID-19 pandemic, there was limited access to hospital inpatients, and patients who could not be evaluated as in the trial protocol were not included.

### 2.4. Study Intervention

The participants were prescribed topical cyclosporine-A 0.1% (Ikervis^®^, Santen Pharmaceuticals Asia Pte Ltd., Osaka, Japan) prophylaxis once daily to both eyes and preservative-free sodium hyaluronate 0.3% (Hialid Mini, Santen Pharmaceuticals Asia Pte Ltd., Osaka, Japan) lubricating eyedrop to be used pro re nata (PRN) from 3 to 5 weeks prior to allo-HSCT to 12 months after allo-HSCT. Based on the transplant protocol used, the participants received systemic GVHD therapy with systemic CsA or tacrolimus combined with a course of methotrexate or mycophenolate. Antithymocyte globulin (ATG) or post-transplant cyclophosphamide was added in cases of unrelated/sibling donor with antigen mismatch or haploidentical transplant, respectively. (Refer to our previous report for the systemic diagnoses and the regimen of systemic immunosuppression in these patients) [11].

### 2.5. Assessment of Tolerance/Compliance with Ikervis^®^

Compliance with the use of Ikervis^®^ in this study was based on the daily drug diary [11]. Briefly, compliance was defined as usage on more than 70% of days and less than five blocks of three consecutive days of not using Ikervis^®^ eyedrops.

### 2.6. Study Visits

Study visits occurred at 3–5 weeks prior to allo-HSCT (screening visit), right before allo-HSCT, and 3 months, 6 months, and 12 months post-allo-HSCT (Appendix A).

### 2.7. Clinical Procedures

Details of the clinical procedures were published elsewhere [11]. Briefly, at each study visit, symptoms were evaluated using the Standardized Evaluation of Eye Dryness (SPEED) questionnaire [18]. Corneal staining was evaluated on a slit lamp biomicroscopy using five zones after the addition of fluorescein dye from a wetted Floret [19,20]. The non-invasive tear breakup times [21,22] and the conjunctival redness grading [23] were performed using the Oculus Keratograph 5M [24]. Tear osmolarity was determined using a TearLab osmometer at the baseline study visit [25,26]. The number of liquid meibum-expressing glands in the lower eyelids was evaluated using a standard force meibomian gland evaluator at baseline and final visit [27].

### 2.8. Tear Cytokines

Schirmer strips from each eye collected over five minutes (without anesthesia) were frozen at −80 Celsius until ready for extraction.

Tear cytokines in all tear samples were analyzed using Olink^®^ Target 96 (Thermo Fisher Scientific, Waltham, MA, USA). Schirmer strips with tear fluid samples were rolled up and transferred to a Spin-X centrifuge tube (0.22 µm Cellulose Acetate; 98231-UT-1; Costar^®^, Costar, Glendale, CA, USA) and submerged in 300 µL of elution buffer, consisting of 0.55 M NaCl (S3014-500G, Sigma Life Science, Darmstadt, Germany), 0.33% Tween 20 (P1379-500ML; Sigma Life Science), and 0.55% Bovine Serum Albumin (9998S; Cell Signalling Technology, Danvers, MA, USA). The samples were subjected to an incubation at 450 rpm for 10 mins at 21 °C. The eluted tears were subsequently centrifuged at 16,000× *g* for 10 min at 21 °C. Subsequently, the clear supernatant was collected and stored in 1.5 mL Protein Lo-bind Tubes (022431081; Eppendorf^®^, Eppendorf, Hamburg, Germany) at −80 °C until the day of analysis.

After the extraction of tears, cytokines were assayed by the Proximity Extension Assay O-linked target 96 platform, a very sensitive method of analysis for a panel of 96 cytokines previously reported [28]. Briefly, each oligonucleotide antibody pair contains unique DNA sequences, allowing hybridization only to each other. Subsequent proximity extension created 96 unique DNA reporter sequences (amplified by real-time PCR).

Concentrations normalized to tear volume, collected as reported in a previous paper, were estimated from the wetting of the Schirmer strip+3 mm.

### 2.9. Statistical Analysis

Only the right eye from each participant was analyzed to avoid the influence of the high correlation between eyes.

The patients were divided into two groups based on the impression flow cytology analysis in our previous report: Group 1 (*n* = 8 conjunctival CD4 cells responding to cyclosporine) and Group 2 (*n* = 5 conjunctival CD4 cells not suppressed after cyclosporine, where patients were non-compliant with cyclosporine). Group 2 also contained one patient who had central cornea staining prior to HSCT and continued to have some signs of DED during the visits despite being compliant with topical cyclosporine [11].

The method of analysis of the repeated measures used linear mixed model regression (Stata), where Y is the cytokine concentration. The procedure used was xtmixed y profile##time || id:, var reml, where Id is the identifier of each patient and Var is the variance of the regression (mixed linear). To obtain joint tests (multiple degrees of freedom) of the interaction and main effects, we used the contrast group##time command. Relevant time trends of specific cytokines were plotted using Stata’s margins and marginsplot commands. More details can be found in the following REF citation: Bruin, J. 2006. newtest: command to compute new test. UCLA Statistical Consulting Group. https://stats.oarc.ucla.edu/stata/ado/analysis/ (accessed on 16 June 2025).

Linear mixed-effect models (LMMs) were used to examine the association between individual clinical parameters and each of the tear cytokine concentrations. The LMMs were adjusted for the correlation between both eyes and repeated time points. Statistical significance of the beta was defined as *p* < 0.05.

## 3. Results

Please refer to our previous paper for the clinical evaluation results of these patients (Appendix A) [11]. Briefly, no participant developed significant ocular signs and symptoms during the study visits, except for one participant who already suffered from central corneal staining prior to the HSCT. The *p*-values for the time effect refer to whether there is a significant alteration of a specific cytokine between different times after HSCT. The *p*-values for the group effect refer to whether there is a difference between the two groups (with respect to cyclosporine usage) overall. The *p*-values for the group*time interaction seek to reveal differences in the tear cytokine level between the two groups at specific times but not at other times.

### 3.1. Clinical Symptom Changes

Across all the participants, the severity of SPEED symptoms was increased when the levels of four cytokines were elevated: fibroblast growth factor (FGF)5 (beta 10.383 [95%CI 2.906–17.861], *p* = 0.006), interleukin (IL)2 (beta 5.897 [95%CI 1.108–10.686], *p* = 0.016), tumor necrosis factor receptor superfamily member (TNFRSF)9 (beta 4.283 [95%CI 1.481–7.084], *p* = 0.003) and glial cell line-derived neurotrophic factor (GDNF) (beta 3.824 [95%CI 0.024–7.623], *p* = 0.049).

The scatter plots for the tear levels of these four cytokines and SPEED symptom scores are shown in Figure 1.

### 3.2. Clinical Sign Changes

Across all the participants, the level of the conjunctival redness varied positively with the level of three of the cytokines: CD8A (beta 0.33 [95%CI 0.02–0.63], *p* = 0.04), IL2RB (beta 0.50 [95%CI 0.05–0.94], *p* < 0.001), and IL33 (beta 0.63 [95%CI 0.02–1.23], *p* = 0.04).

Increased levels of IL17A were associated with increased cornea staining (beta 0.387 [95%CI 1.301–6.334], *p* = 0.003).

Increased levels of one cytokine, FGF23, were associated with reduced expressivity of meibomian glands (number of liquid meibum expressed from lower eyelids), i.e., FGF23 (beta −2.259 [95%CI −4.5 to −0.018], *p* = 0.048), whereas IL20 was associated with increased expressivity (beta 1.999 [95%CI 0.043–3.955], *p* = 0.045).

Most interestingly, the tear level of fourteen cytokines was associated with baseline tear osmolarity. Nine cytokines increased with tear osmolarity levels, whereas five cytokines were inversely correlated with tear osmolarity (Table 1). Increased levels of matrix metalloproteinase (MMP) 1 and certain pro-inflammatory cytokines (CCL3, CCL4, IL 1alpha, and IL12beta) were associated with increased tear osmolarity during the study visits.

### 3.3. Tear Cytokine Changes

The tear concentration of 38 cytokines (Table 2) showed a significant change (*p* < 0.05) over time, and for many of these, the elevation was marked at the one-year visit. These include gamma-interferon (IFNG), CXC-chemokine ligand (CXCL)9, CC-chemokine ligand (CCL)3, and CCL4 (all *p* < 0.0001) (Figure 2).

For gamma-interferon, there was significant interaction between group and time at the 1-year visit (*p* = 0.022), where the cytokine was significantly suppressed in Group 1. Four other cytokines showed significant group and time interaction at the last (1 year) visit: FGF23 (*p* = 0.027), FGF5 (*p* = 0.035), leukemia inhibitory factor receptor (LIFR) (*p* = 0.044), and advanced glycation end-products binding protein (ENRAGE) (*p* = 0.036). Tear FGF23 and FGF5 were suppressed by cyclosporine, whereas LIFR and ENRAGE were depressed by cyclosporine (Figure 3).

Prior to HSCT (first two visits), T cell stimulant tumor necrosis factor superfamily (TNFSF) 14 was suppressed in Group 1 (*p* = 0.039 by paired T test) but not in Group 2 (Figure 4).

Systemic immunosuppression, as decided by the hematologists managing the patients, tended to be reduced or withdrawn at the end of six months in all the participants (Table 3 and Table 4).

## 4. Discussion

This study shows that HSCT induces a significant elevation of 38 tear cytokines/chemokines, which becomes most marked at the one-year visit, even without the occurrence of ocular graft-versus-host disease. The reduction in or stoppage of the systemic immunosuppression at six months occurred at the same time as the increase in tear cytokines.

The findings of this study suggest that an elevation in tear cytokines is a phenomenon that may occur after HSCT, even though no clinical ocular GVHD occurred up to one year. It remains to be seen if tear cytokines would increase if systemic immunosuppression had not been followed in the first year. Given that the tear cytokines increased, it is unclear whether ocular GVHD will occur in these patients years or sometime after the cessation of the topical cyclosporine. These patients will continue to be monitored by the study team. This has implications for the pathophysiology of chronic ocular GVHD. It may be that immune cells or progenitor cells migrate or traffic to ocular glands and the ocular surface over a long time after HSCT.

Our study also verified that changes in the levels of tear cytokines occurred with changes in the clinical parameters of dry eye disease. This highlights the real-world significance of cytokine levels.

The fact that many cytokines increased between six months and one year after transplantation, corresponding to the period when most of the systemic immunosuppression was reduced or stopped, suggests that systemic immunosuppression may be critical for maintaining an anti-inflammatory milieu in the tear and ocular surface. Beyond the one-year period, topical cyclosporine on its own may not be sufficient to prevent further elevation in pro-inflammatory cytokines.

Cyclosporine works by inhibiting calcineurin/nuclear factor of activated T cell signaling, inducing apoptosis of T cells [29], and inhibiting immunoglobulin synthesis [30]. There are plenty of T cells in the human conjunctiva [31,32,33,34,35]. It has been shown that treatment with topical cyclosporine resulted in a reduction in T cells in the conjunctiva epithelium and stroma of people with Sjogren’s syndrome [36,37] and in people with allergic conjunctivitis [38].

IFNG is a well-known cytokine in DED, which drives the inflammatory response in the ocular surface. It is interesting that topical cyclosporine suppresses the rise in this cytokine at the one-year time point [39,40,41,42,43] and that the participants compliant with topical cyclosporine had a significant reduction in conjunctival CD4 T cells [11].

The FGF family has been implicated in multiple biological processes in mammals, such as cell proliferation and differentiation, including tissue repair. The functions of LIFR and ENRAGE in the post-HSCT context are unknown.

A previous study involving thirty healthy volunteers found a link between increased cytokines and increased tear osmolarity [44]. Tear cytokines have also been linked to the severity of dry eye disease in patients [45].

A cross-sectional study of patients with dry eye disease also reported a link between elevated IFNG and elevated tear osmolarity [42].

An interesting in vitro study on human corneal epithelial cells and a clinical study of patients with dry eye disease found an association between elevated IL33 and various parameters of dry eye disease (positive correlation with OSDI and corneal staining; negative correlation with TBUT). However, the study did not evaluate conjunctival redness. The authors suggested that the IL-33/ST2 pathway regulates ocular surface inflammation [46].

A previous study described the importance of tear cytokines in dry eye disease [47]. A study showed lowered levels of IL-1β, IL-6, and TNF-α in dry eye participants after treatment with topical cyclosporine tds [48]. Our study differs in findings because of the differences in study subjects (our study participants were HSCT patients without DED diagnosis) and the treatment formulation and regimen.

The levels of tear cytokines changed after treatment of meibomian gland dysfunction with intense pulse light, thus supporting our findings on changes in gland expressivity and cytokines [49].

We do not know why topical cyclosporine reduced tear TNFSF14 in some participants and not others prior to HSCT. More research is necessary to determine if this pre-HSCT response of TNFSF14 can predict the ability of cyclosporine to lower other tear cytokines post-HSCT.

One of the clinical implications of this study is that novel treatments with serine protease inhibitors may be possible in dry eye disease, and tear cytokine levels may be used as markers of response to treatment [2,50,51].

The strength of this study is that for all the participants, tear samples were available and analyzable at all the time points with no missing data. The availability of flow cytometric data on T cells is also an advantage because it makes the interpretation of cytokines, like IFNG, easier. An increase in IFNG corresponds to and is explainable by the observed increase in CD4 conjunctival T cells in our previous report. The suppression of the IFNG rise corresponds to participants whose CD4 T cells did not increase in the conjunctiva.

The limitation of this study is that it has a small sample size and was terminated at one year after HSCT. Previous studies have found ocular GVHD to occur many years after HSCT. The difficulty in studying patients many years after HSCT is that the DED from ocular GVHD may be indistinguishable clinically from idiopathic, environmentally induced, or age-related DED. This study was not designed to evaluate the use of topical cyclosporine versus a comparator drop, such as sodium hyaluronate. Therefore, caution should be exercised when interpreting the results concerning the two group comparisons. Although there is no universally agreed-upon best method of tear collection, the method using Schirmer’s strip strikes a good compromise between ease of collection and other factors, such as protein yield and potential for reflex tearing, and was employed successfully in our previous tear cytokine studies.

Our study shows that topical daily cyclosporine eyedrops can suppress some inflammatory cytokines, e.g., tear IFNG. It remains to be seen whether the suppression of specific cytokines up to one year can have more lasting prophylaxis beyond that time. It is neither feasible nor practical to continue the administration of cyclosporine in asymptomatic people who had HSCT beyond that time frame. The management of ocular complications of GVHD should be multidisciplinary, involving systemic immunosuppression.

## 5. Conclusions

In conclusion, we are able to quantify longitudinal changes in tear cytokines after HSCT using a sensitive method. These have a tendency to increase long after the transplant, corresponding to the period of reduced systemic immunosuppression, and prophylactic cyclosporine eyedrops may suppress inflammatory cytokines, like IFNG. The association of an elevation in tear cytokine levels with various clinical parameters of dry eye suggests that these may serve as markers of ocular surface inflammation.

## Figures and Tables

**Figure 1 ijms-26-05915-f001:**
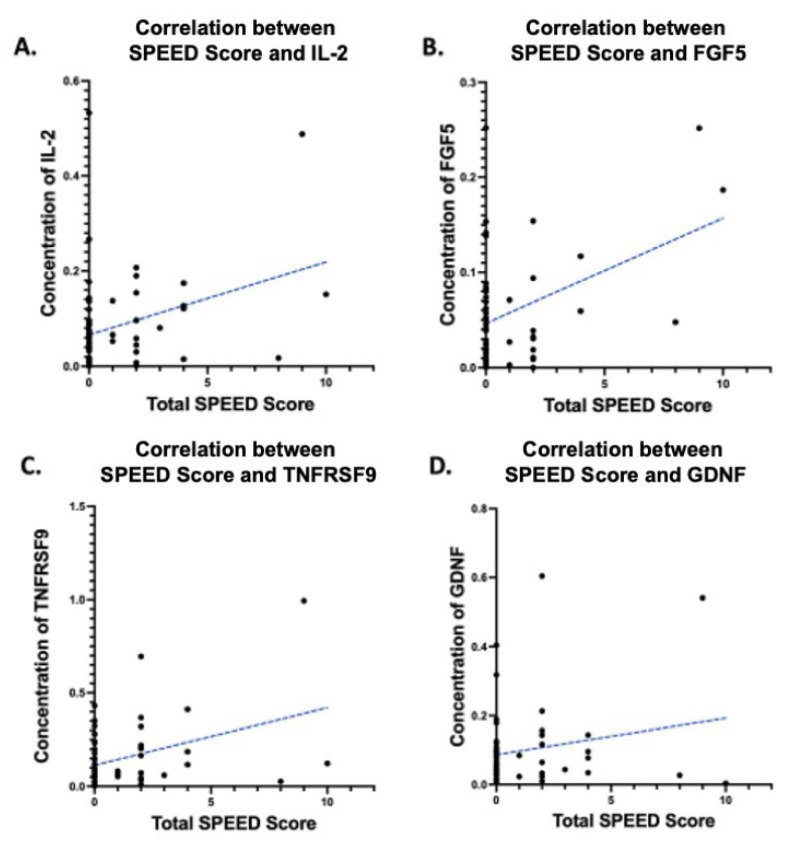
Scatter plots showing cytokines that are significantly associated with standard patient evaluation of eye dryness (SPEED) scores in multivariate analyses. (**A**). Interleukin (IL)-2, (**B**). fibroblast growth factor (FGF)-5, (**C**). tumor necrosis factor receptor superfamily member (TNFRSF)-9, and (**D**). glial cell line-derived growth factor (GDNF). The dashed line represents the regression line.

**Figure 2 ijms-26-05915-f002:**
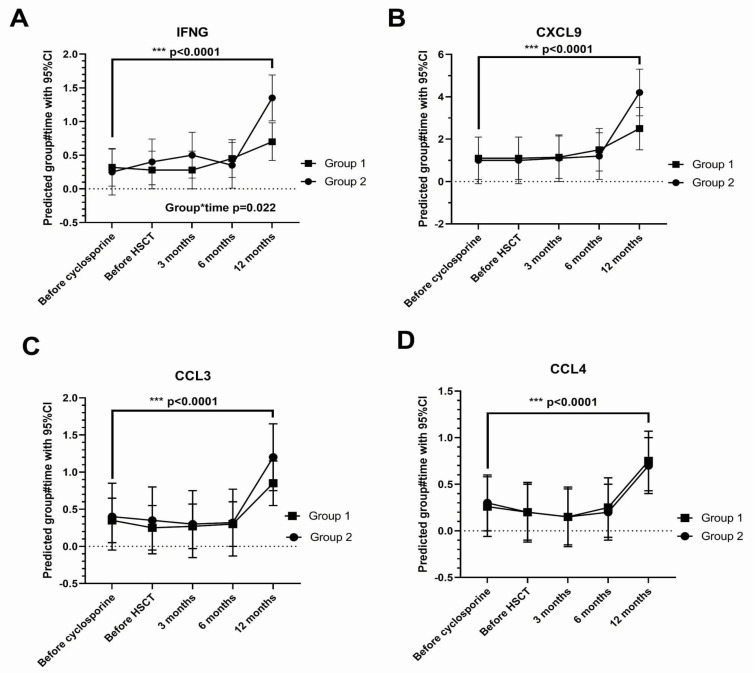
Cytokine changes over time. (**A**) (interferon gamma [IFNG]; (**B**) chemokine (C-X-C motif) ligand [CXCL9]; (**C**) chemokine (C-C motif) ligand [CCL3], and (**D**) CCL4). Drug-responsive Group 1 (*n* = 8), non-responsive Group 2 (*n* = 5). Group*time interaction not statistically significant (*p* > 0.05) except at the time point indicated. Error bars: 95% confidence intervals. The dashed line represents zero. *** *p* < 0.001.

**Figure 3 ijms-26-05915-f003:**
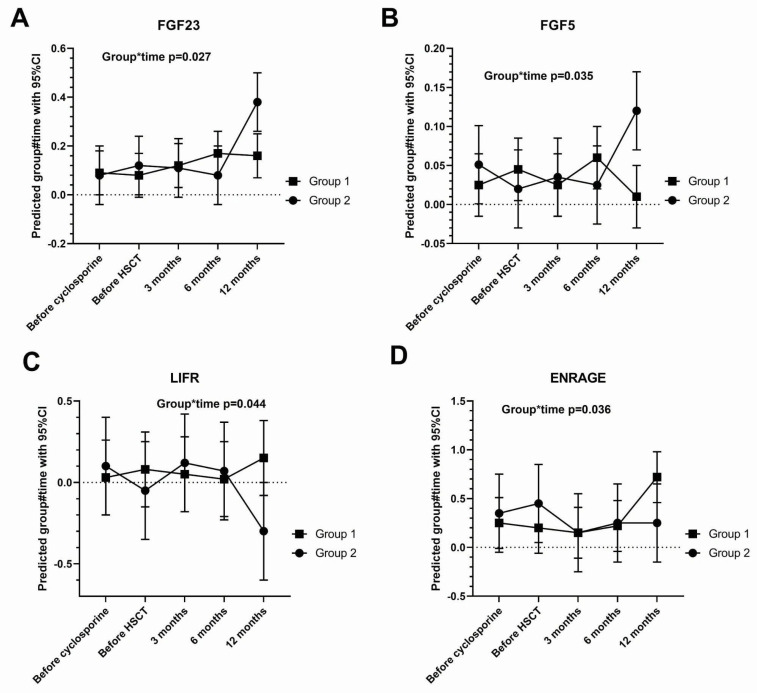
Cytokine changes with significant group*time interaction: Interferon gamma [IFNG] (shown in Figure 1). (**A**). Fibroblast growth factor [FGF23], (**B**). FGF5, (**C**). leukemia inhibitory factor receptor [LIFR], and (**D**). advanced glycation end-products binding protein [ENRAGE]. Drug-responsive Group 1 (*n* = 8); non-responsive Group 2 (*n* = 5). Group*time interaction not statistically significant (*p* > 0.05) except at the time point indicated. Error bars: 95% confidence intervals. The dashed line represents zero.

**Figure 4 ijms-26-05915-f004:**
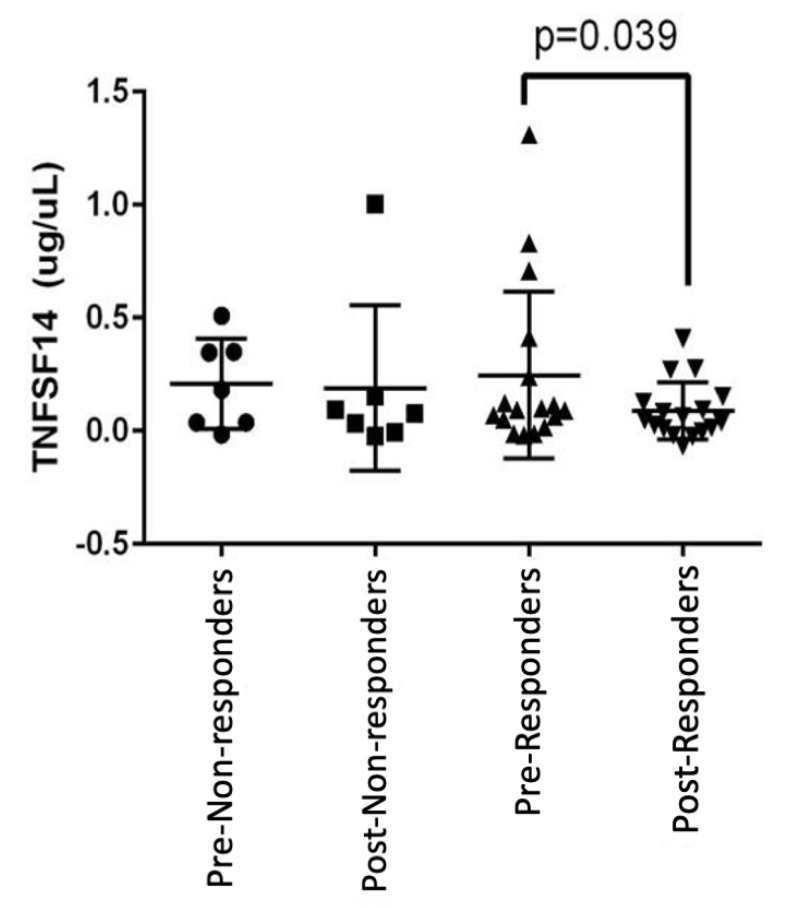
Dot plots showing the concentrations of TNFSF14 in the four groups. Horizontal line: group means; error bars: 95% confidence intervals. *p*-values (via paired T test comparing before [pre] and after treatment [post] with cyclosporine at 2–4 weeks, prior to HSCT).

**Table 1 ijms-26-05915-t001:** Relationship between changes in tear osmolarity and tear cytokine levels.

Interleukin	Beta (Coefficient)	*p*-Value	95% Confidence Interval (CI)	
			Lower End	Upper End
BETANGF	281.7	0.04	19.71	543.69
IL12B	83.95	0.01	24.33	143.57
TRANCE	64.58	0.04	4.07	125.09
TNFSF14	32.96	0	11.32	54.6
CCL3	24.51	0.01	7.26	41.77
CCL4	23.07	0.01	5.45	40.68
IL1α	20.24	0.01	5.95	34.53
TGFα	13.62	0.03	1.17	26.07
MMP1	5.83	0.01	1.33	10.33
CCL11	−77.52	0.02	−140.51	−14.53
SLAMF1	−71.25	0.02	−132.85	−9.65
IL7	−62.41	0.02	−112.88	−11.93
IL10RA	−46.35	0.03	−87.58	−5.12

**Table 2 ijms-26-05915-t002:** *p*-values of cytokines with a significant time trend.

Cytokine	*p*-Value (from Linear Mixed Model Regression)
CXCL9	<0.0001
CCL4	<0.0001
CCL3	<0.0001
IFN-gamma	<0.0001
CXCL11	0.0004
CXCL10	0.0005
TRANCE	0.0014
ADA	0.0018
IL12B	0.0022
FGF23	0.0024
CD5	0.0024
IL2	0.0037
IL2RB	0.0042
Trail	0.0068
NRTN	0.0069
TNFSF9	0.0072
UPA	0.0083
TGFAlpha	0.0086
TNFSF14	0.0106
IL18	0.0115
IL18R1	0.0122
IL22RA1	0.0133
IL8	0.0176
CDCP1	0.0182
CD40	0.0183
IL20	0.0219
SIRT2	0.0252
HGF	0.0261
CASP8	0.0275
CSF1	0.0285
IL6	0.0304
STAMBP	0.0308
CCL20	0.0312
EBP1	0.032
CXC11	0.0336
FLT31	0.038
FGF19	0.0455
IL1A	0.0493

**Table 3 ijms-26-05915-t003:** Systemic drug used by participants in Group 1.

Participant Number	Systemic Drug	Dosage		
		After HSCT	3 Months Post-HSCT	6 Months Post-HSCT
1	Predisolone	-	-	5 mg (EOD, 2×/wk)
	Mycophenolic acid EC (Myfortic)	720 mg, 360 mg	180 mg	-
	Tacrolimus	nil	-	-
	Ciclosporin (NEORAL)	150 mg, 125 mg, 175 mg, 125 mg, 75 mg, 10 mg, 3 mg	50 mg, 75 mg, 50 mg	40 mg, 25 mg
2	Tacrolimus	1 mg	-	-
3	Ciclosporin (NEORAL)	125 mg	75 mg, 50 mg	-
4	Mycophenolic acid EC (Myfortic)	720 mg, 540 mg, 360 mg	-	-
	Ciclosporin (NEORAL)	200 mg, 225 mg, 125 mg	50 mg, 25 mg	-
5	Predisolone	50 mg, 40 mg	10 mg, 25 mg	-
	Mycophenolic acid EC (Myfortic)	720 mg	-	-
	Tacrolimus	1 mg	0.5 mg (4×/wk), 0.2 mg (2×/wk)	-
6	Dexamethasone	5 mg	4 mg	-
	Mycophenolic acid EC (Myfortic)	720 mg	-	-
	Ciclosporin (NEORAL)	150 mg, 50 mg, 100 mg	75 mg	-
7	Mycophenolic acid EC (Myfortic)	360 mg	180 mg, 360 mg	-
	Ciclosporin (NEORAL)	250 mg, 100 mg, 25 mg	-	-
8	Mycophenolic acid EC (Myfortic)	720 mg	-	-
	Ciclosporin (NEORAL)	175 mg, 50 mg	50 mg, 25 mg, 25 mg (3×/wk)	-

Group 1 (*n* = 8 conjunctival CD4 cells responding to cyclosporine).

**Table 4 ijms-26-05915-t004:** Systemic drug used by participants in Group 2.

Participant Number	Systemic Drug	Dosage		
		After HSCT	3 Months Post-HSCT	6 Months Post-HSCT
9	Mycophenolic acid EC (Myfortic)	-	720 mg, 540 mg	360 mg
	Tacrolimus	1 mg	-	-
10	Predisolone	60 mg, 20 mg, 15 mg, 10 mg	10 mg, 5 mg, 5 mg (EOD)	10 mg, 5 mg
	Mycophenolic acid EC (Myfortic)	360 mg, 180 mg, 180 mg EOD, 180 mg (2·/week)	-	-
	Tacrolimus	2.5 mg, 3 mg, 2 mg, 0.5 mg, 1.5 mg	1 mg, 0.5 mg	-
11	Predisolone	30 mg, 20 mg, 10 mg	-	10 mg, 5 mg
	Tacrolimus	2 mg, 1.5 mg	1 mg, 0.5 mg (3×/wk), 0.5 mg (2×/wk)	-
12	Tacrolimus	1.5 mg	-	-
13	Tacrolimus	1.5 mg, 1 mg	0.5 mg	-

Group 2 (*n* = 5 conjunctival CD4 cells not suppressed after cyclosporine, where patients were non-compliant with cyclosporine).

## Data Availability

The datasets generated during and/or analyzed during the current study are not publicly available as the informed consent signed by the study subjects did not include public sharing of their data.

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
