# Peer review of "Tear Cytokine Changes up to One Year After Allogeneic Hematopoietic Stem Cell Transplant: Effect of Daily Topical Cyclosporine-A 0.1% Emulsion"

_ijms, 2025, doi:10.3390/ijms26125915_

Round 1
Reviewer 1 Report
Comments and Suggestions for Authors
The study aims at profiling cytokine changes in patients who underwent HSCT and were treated prophylactically with topical CSA 0.1%. the study is part of a study that was published in ophtalmol Ther in 2023 showing a decreased proportion of conjunctival CD4 T cells in patients compliant to treatment with topical CSA.
The study shows a significant elevation of 38 cytokines over time and found several of them to be associated with changes in tear osmolarity or symptom score.
As much as the topic is interesting and important in order to understand the mechanism and treatment options for DES in patients undergoing HSCT the manuscript uses a very small sample of patients divided to two groups according to compliance adherence and CD4 count. More over adequate control groups (such as healthy individuals or patients with DES not related to transplants or even patients receiving sodium hyaluronate lubricating eyedrops with no CSA) are missing. The method chosen by the researchers to analyze the samples should deal with the protein recovery efficiency, protein loss in the process, normalization challenges and the potential for reflex tearing changing tear composition.
The results are presented in a cumbersome manner and the relevant statistical differences are not completely clear.
Moreover the clinical importance of this results are even more not clear, since only one patient developed DES GVHD related in the original manuscript and again no relevant control was tested
The discussion lacks a clear presentation of what is known in the literature on the subject and how this manuscript adds to this while jumping to unsupported conclusions. More over there is a manuscript by wu et al from 2024 dealing with analyzing interleukin-1β, interleukin-6, and tumor necrosis factor-α levels in dry eye and the therapeutic effect of cyclosporine A [PMID: 39247746 DOI: 10.12998/wjcc.v12.i25.5665], which was not mentioned in the manuscript
Author Response
We would like to thank the reviewers for their time and effort in reviewing the paper. We would like to submit a revised manuscript and address the comments below.
Reviewer 1:
The study aims at profiling cytokine changes in patients who underwent HSCT and were treated prophylactically with topical CSA 0.1%. the study is part of a study that was published in ophtalmol Ther in 2023 showing a decreased proportion of conjunctival CD4 T cells in patients compliant to treatment with topical CSA.
The study shows a significant elevation of 38 cytokines over time and found several of them to be associated with changes in tear osmolarity or symptom score.
- As much as the topic is interesting and important in order to understand the mechanism and treatment options for DES in patients undergoing HSCT the manuscript uses a very small sample of patients divided to two groups according to compliance adherence and CD4 count. More over adequate control groups (such as healthy individuals or patients with DES not related to transplants or even patients receiving sodium hyaluronate lubricating eyedrops with no CSA) are missing.
Response:
This is not a study designed to compare tear cytokines between cross sectional groups such as DES and healthy individuals. As DED is multifactorial, such comparisons would indeed require large samples. We are only interested in longitudinal changes in tear cytokines, comparing each participant’s changes with their own levels before the HSCT, which reduces the sample size since inter subject variation is not a concern. Performing a multipoint cytokine evaluation in a large group would be very expensive and might not be considered cost effective by grant funders. We agree that the two groups comparison is essentially post-hoc in nature, since we started only with a single observational group. This two group classification is only meaningful to look at the effect of using topical cyclosporine consistently, but is not required in the time after HSCT analysis of the overall group.
In usual hospital practice, hematologists would only refer HSCT patients to eye departments should ocular symptoms or signs arise. Therefore, there is value in capturing consecutive cases of HSCT regardless of ocular complaints, to study their evolution of mucosal immune response, if any. There is a need to observe and understand HSCT induced changes, regardless of whether prophylactic Ikervis was used (which is a separate question).
We added this sentence to the discussion:
“The study wasn’t designed to evaluate use of topical cyclosporine versus using a comparator drop such as sodium hyaluronate. Therefore caution should be exercised when interpreting the results concerning the two group comparisons.”
- The method chosen by the researchers to analyze the samples should deal with the protein recovery efficiency, protein loss in the process, normalization challenges and the potential for reflex tearing changing tear composition.
Response:
In previous studies, the protein recovery and loss, as well as normalization using Schirmer strips to collect tear, have been investigated. This method is now increasingly used, compared to other methods of tear collection such as microcapillary tubes. Some examples are:
Koduri MA, Prasad D, Pingali T, Singh VK, Shanbhag SS, Basu S, Singh V. Optimization and evaluation of tear protein elution from Schirmer's strips in dry eye disease. Indian J Ophthalmol. 2023 Apr;71(4):1413-1419. doi: 10.4103/IJO.IJO_2774_22. PMID: 37026274; PMCID: PMC10276694.
Denisin AK, Karns K, Herr AE. Post-collection processing of Schirmer strip-collected human tear fluid impacts protein content. Analyst. 2012 Nov 7;137(21):5088-96. doi: 10.1039/c2an35821b. Epub 2012 Sep 18. PMID: 22991688.
Gijs M, Arumugam S, van de Sande N, Webers CAB, Sethu S, Ghosh A, Shetty R, Vehof J, Nuijts RMMA. Pre-analytical sample handling effects on tear fluid protein levels. Sci Rep. 2023 Jan 24;13(1):1317. doi: 10.1038/s41598-023-28363-z. PMID: 36693949; PMCID: PMC9873914.
Pieczyński J, Szulc U, Harazna J, Szulc A, Kiewisz J. Tear fluid collection methods: Review of current techniques. Eur J Ophthalmol. 2021 Sep;31(5):2245-2251. doi: 10.1177/1120672121998922. Epub 2021 Feb 25. PMID: 33631970.
These issues have also been discussed by recent presentations from the international Tear Protein Network at the 26th International Ocular Surface Society meeting May 3, 2025. Eg, “Standardizing tear fluid biomarker research: A comprehensive review of collection and analysis protocols” by Marlies Gijs and “Potential normalizing proteins in tear fluid” by Ashok Sharma.
We added this sentence to the discussion:
“Although there is no universally agreed best method of tear collection, the method using Schirmer’s strip strikes a good compromise between ease of collection and other factors such as protein yield and potential for reflex tearing, and has been employed successfully in our previous tear cytokine studies.”
We cite these as examples:
Tong L, Wong TY, Cheng Y. Level of tear cytokines in population-level participants and correlation with clinical features. Cytokine. 2018 Oct;110:452-458. doi: 10.1016/j.cyto.2018.05.013. Epub 2018 May 24. PMID: 29803660.
Willems B, Tong L, Minh TDT, Pham ND, Nguyen XH, Zumbansen M. Novel Cytokine Multiplex Assay for Tear Fluid Analysis in Sjogren's Syndrome. Ocul Immunol Inflamm. 2021 Nov 17;29(7-8):1639-1644. doi: 10.1080/09273948.2020.1767792. Epub 2020 Jul 13. PMID: 32657632.
- The results are presented in a cumbersome manner and the relevant statistical differences are not completely clear.
Response:
We’ve used a well established method of linear mixed model regression analysis on the Stata software. To clarify the statistical differences, we added these sentences to the results:
“The p values for time effect refer to whether there is a significant alteration of a specific cytokine between different times after HSCT. The p values for group effect refer to whether there is a difference between the two groups (with respect to cyclosporine usage) overall. The p values for group* time interaction seek to reveal differences of a tear cytokine level between the two groups at specific times but not at other times.”
- Moreover the clinical importance of this results are even more not clear, since only one patient developed DES GVHD related in the original manuscript and again no relevant control was tested.
Response:
We agree that these results alone are not enough to support the use of cyclosporine for HSCT. Based on the results of this study alone, we do not recommend the use of prophylactic cyclosporine, since one year is not long enough for participants to develop GVHD.
Continual repeat assessment of cytokines beyond one year may also not be logistically feasible (see response to point 1). As explained in our response to point 1, we do need to understand what happens after HSCT. In this report, we are just pointing out the clear trend of changes of these cytokines with respect to time after HSCT, ie, the increase of many cytokines between 6 months to one year, which will need further investigation because it may have significant implication on the survival of immune cells in the ocular surface beyond one year of HSCT. This issue needs further discussion with hematologists as it may require systemic immunosuppression beyond what is commonly used.
- The discussion lacks a clear presentation of what is known in the literature on the subject and how this manuscript adds to this while jumping to unsupported conclusions. More over there is a manuscript by wu et al from 2024 dealing with analyzing interleukin-1β, interleukin-6, and tumor necrosis factor-α levels in dry eye and the therapeutic effect of cyclosporine A [PMID: 39247746 DOI: 10.12998/wjcc.v12.i25.5665], which was not mentioned in the manuscript
Response:
We thank the reviewer for mentioning Wu et al’s study.
There is indeed a number of papers on tear cytokine evaluation in DED (one or two from our group), but we hesitate to bring the discussion in this direction because it will increase the length of the discussion considerably and since our participants do not have DED in this period, the literature may be of marginal relevance.
In Wu et al, compared to the comparison group which used only artificial tears, the group of dry eye participants treated with additional topical cycloporine 0.05% tds eyedrops for eight weeks resulted in significantly lowered levels of tear cytokines IL-1β, IL-6, and TNF-α. We added this statement to the discussion:
“A study has shown lowered levels of IL-1β, IL-6, and TNF-α in dry eye participants after treatment with topical cyclosporine tds.48 Our study differs in findings because of the differences in study subjects (our study participants were HSCT patients without DED diagnosis) and the treatment formulation and regime.”
Citation added: 48. Wu J, Li GJ, Niu J, Wen F, Han L. Analyze interleukin-1β, interleukin-6, and tumor necrosis factor-α levels in dry eye and the therapeutic effect of cyclosporine A. World J Clin Cases. 2024 Sep 6;12(25):5665-5672. doi: 10.12998/wjcc.v12.i25.5665. PMID: 39247746; PMCID: PMC11263063.
We also added this sentence in the discussion:
“A previous study has described the importance of tear cytokines in dry eye disease.47”
The citation is:
47Roda M, Corazza I, Bacchi Reggiani ML, Pellegrini M, Taroni L, Giannaccare G, Versura P. Dry Eye Disease and Tear Cytokine Levels-A Meta-Analysis. Int J Mol Sci. 2020 Apr 28;21(9):3111. doi: 10.3390/ijms21093111. PMID: 32354090; PMCID: PMC7246678.

Reviewer 2 Report
Comments and Suggestions for Authors
Comments to the Authors
The authors present a manuscript in the form of a brief report related to the effect of cyclosporine emulsion at 0.1% for ophthalmic application in patients as an important treatment factor. This manuscript is based on the evaluation of changes in pro-inflammatory cytokine profiles, compared to other study variables. This is supported by a prospective model in a cohort of patients with tear samples, suggesting that prophylaxis with cyclosporine emulsion suppresses the inflammatory marker IFN-γ (IFN-γ). In the current submission, the authors should review several significant points in their manuscript that are confusing. They also present spelling errors throughout the text and poorly distributed figures that make it difficult to follow the manuscript. Among other comments, I would like to express them according to the sections.
- In their abstract, the authors should include the cytokine levels in lines #27 and 28. Is it unclear whether the values ​​are lower or higher in the presentation? Because they have cytokine and chemokine-type cytokine levels, I suggest rewording this text?
- In the conclusion of the abstract, the same comment is made as above: which cytokines are elevated? Reword lines #32-33.
- In the introduction section, lines #66-70, the authors' objective is unclear, and they mark it in two places from a previous work reported in their reference 11. I suggest rewording it.
- Methods
In the study population section, lines #81 to 84, correct the spaces in your link.
- In the eligibility section, include inclusion and exclusion criteria for the study patient group? 6.- In the intervention of the study line #91 change ikervis to Ikervis ® review the entire text lines # 91,101, 102, 104
7.- In line # 99 to 100, should the authors rephrase this paragraph, according to the data obtained in reference 11. In the current form, it only says to consult the reference? 8. In clinical procedures, show the meaning of the acronyms in the SPEED questionnaire, line # 110 (Symptom, Progression, and Efficacy of Dry Eye Disease).
- In tear cytokines, line #, include data from the freezing equipment. Review the spaces in line #122: 300 ul to µL, spaces after mins, line #125, 127 mL.
- In the statistical analysis section, lines # 136-158, explain in more detail why the analysis is only for one eye. Also, correct the spaces in the text on lines #151-150.
- In the results section, change the beginning of the description of the results from its current form. Can't you say that the reference is consulted? It is the description of the study data. I suggest rephrasing the text on lines 159-163.
- Standardize your figures in the legend, whether they are bold or not, and check if you had problems formatting your manuscript because they overlap. I suggest carefully reviewing all figures, lines # 170 to 175.
- Standardize the nomenclature of cytokines in the text and in the graphs or tables presented by the authors, lines # 190-193.
- In Table 1, the authors include colors in yellow, orange, and blue, lines # 192 to 193. Include an explanation of these color labels at the end of the table.
- I suggest increasing the sharpness and resolution of Figure 2 to highlight its data, lines # 197 to 198.
- Correct the font size, lines # 207.
- Structure your Table 2, lines # 211 to 216.
- Review the description of your graphs. The authors may have had problems uploading their manuscript and medication table, lines # 234 to 236.
- I suggest the authors expand on their study limitations and include some perspectives in the conclusion. Future
20.- I suggest the authors reinforce this paragraph of the discussion line # 295 to 296
Comments on the Quality of English Language
The authors could improve the English of their manuscript to improve the description of their data.
Author Response
Reviewer 2
Comments to the Authors
The authors present a manuscript in the form of a brief report related to the effect of cyclosporine emulsion at 0.1% for ophthalmic application in patients as an important treatment factor. This manuscript is based on the evaluation of changes in pro-inflammatory cytokine profiles, compared to other study variables. This is supported by a prospective model in a cohort of patients with tear samples, suggesting that prophylaxis with cyclosporine emulsion suppresses the inflammatory marker IFN-γ (IFN-γ).
In the current submission, the authors should review several significant points in their manuscript that are confusing. They also present spelling errors throughout the text and poorly distributed figures that make it difficult to follow the manuscript. Among other comments, I would like to express them according to the sections.
Response:
We performed a search for ‘spelling errors’, and reviewed the distribution of figures according to the explanation of results (see response to comment 3 reviewer 1). We hope the following revisions will clarify the manuscript and make it less confusing.
- In their abstract, the authors should include the cytokine levels in lines #27 and 28. Is it unclear whether the values are lower or higher in the presentation? Because they have cytokine and chemokine-type cytokine levels, I suggest rewording this text?
Response:
We referred to chemokines as a subfamily of cytokines. Nevertheless, we reworded this sentence: “Levels of 38 cytokines showed significant changes (p<0.05) over time; and in many; elevation was marked at one year.” to
“Levels of 38 cytokines/chemokines showed significant increase (p<0.05) over time; and in many; elevation was marked at one year.”
- In the conclusion of the abstract, the same comment is made as above: which cytokines are elevated? Reword lines #32-33.
Response:
Thank you. As it is too wordy to mention all the 38 cytokines/chemokines that became elevated, we refer the reader to the main text for this. We reword the sentence to:
“HSCT induces significant elevation of 38 tear cytokines/chemokines…”
- In the introduction section, lines #66-70, the authors' objective is unclear, and they mark it in two places from a previous work reported in their reference 11. I suggest rewording it.
Response:
Thank you. We moved the reference 11 for this sentence:
“Previously, we showed that people who underwent HSCT displayed one of two patterns of change of conjunctival CD4 T-cells, depending on their compliance to the cyclosporine.11 “
to before the sentence:
“Since cytokines in the tear drive ocular surface inflammation, we are interested to determine whether…”
In this way we hope that the objective is clear and is meant to follow up the analysis of a previously reported
group of patients.
Methods
- In the study population section, lines #81 to 84, correct the spaces in your link.
Response:
Thank you.
- In the eligibility section, include inclusion and exclusion criteria for the study patient group?
Response:
We have reworded the inclusion and exclusion criteria, and included the fact that many patients could not be recruited due to COVID-19 regulations.
6.- In the intervention of the study line #91 change ikervis to Ikervis ® review the entire text lines # 91,101, 102, 104
Response:
We performed a search for the word ‘Ikervis’ and replaced with “Ikervis ®".
7 .- In line # 99 to 100, should the authors rephrase this paragraph, according to the data obtained in reference 11. In the current form, it only says to consult the reference?
Response:
This statement refers to the list of systemic disease encountered in the patients, which is already lengthy and already published in our previous report.
- In clinical procedures, show the meaning of the acronyms in the SPEED questionnaire, line # 110 (Symptom, Progression, and Efficacy of Dry Eye Disease).
Response:
We spelled out the acronym SPEED.
9.In tear cytokines, line #, include data from the freezing equipment. Review the spaces in line #122: 300 ul to µL, spaces after mins, line #125, 127 mL.
Response:
We ensured that there is a space between the unit and the number.
- In the statistical analysis section, lines # 136-158, explain in more detail why the analysis is only for one eye. Also, correct the spaces in the text on lines #151-150.
Response:
We added this sentence: “The Schirmer strip for the other eye is reserved for a different type of analysis (mass spectrometry for other tear proteins apart from cytokines) which will be reported in another paper.”
- In the results section, change the beginning of the description of the results from its current form. Can't you say that the reference is consulted? It is the description of the study data. I suggest rephrasing the text on lines 159-163.
Response:
We altered the sentence to “The details of the clinical features of the participants have been reported previously. In summary, …”
- Standardize your figures in the legend, whether they are bold or not, and check if you had problems formatting your manuscript because they overlap. I suggest carefully reviewing all figures, lines # 170 to 175.
Response:
We ensured that only the Figure label is bold and not the title.
- Standardize the nomenclature of cytokines in the text and in the graphs or tables presented by the authors, lines # 190-193.
Response:
We spelled out in full the first time a cytokine name appears in the manuscript main text or in the figure legend. eg, Glial cell line-derived neurotrophic factor (GDNF), but for simplicity, left out the full names in Tables 1 and 2.
- In Table 1, the authors include colors in yellow, orange, and blue, lines # 192 to 193. Include an explanation of these color labels at the end of the table.
Response:
We have uploaded a version of Table 1 without the colors.
- I suggest increasing the sharpness and resolution of Figure 2 to highlight its data, lines # 197 to 198.
Response:
We have uploaded another version of the Figure 2 and 3 generated using Prism Graphpad.
- Correct the font size, lines # 207.
Response:
We have changed the font size.
- Structure your Table 2, lines # 211 to 216.
Response: We have revised the table.
- Review the description of your graphs. The authors may have had problems uploading their manuscript and medication table, lines # 234 to 236.
Response:
We have reformatted Table 3 so that it starts from the left edge like the other tables.
- I suggest the authors expand on their study limitations and include some perspectives in the conclusion. Future
Response:
We added the following lines to the study limitations:
“The study wasn’t designed to evaluate use of topical cyclosporine versus using a comparator drop such as sodium hyaluronate. Therefore caution should be exercised when interpreting the results concerning the two group comparisons.”
and
“Although there is no universally agreed best method of tear collection, the method using Schirmer’s strip strikes a good compromise between ease of collection and other factors such as protein yield and potential for reflex tearing, and has been employed successfully in our previous tear cytokine studies.”
17.- I suggest the authors reinforce this paragraph of the discussion line # 295 to 296
Response:
We added to this: “The strength of the study is that for all these participants, tear samples were available and analyzable at all the time points with no missing data. The availability of flow cytometric data on the T cells is also an advantage."
So that the section now reads:
“The strength of the study is that for all these participants, tear samples were available and analyzable at all the time points with no missing data. The availability of flow cytometric data on the T cells is also an advantage because it makes the interpretation of the cytokines like IFNG easier. An increase in IFNG corresponds to, and is explainable by the observed increase in CD4 conjunctival T cells in our previous report. The suppression of the IFNG rise corresponds to participants whose CD4 T cells did not increase in the conjunctiva."
Comments on the Quality of English Language
- The authors could improve the English of their manuscript to improve the description of their data.
Response:
The descriptions have been reworded as above and also where there is similarity with the previous manuscript (highlighted).

Round 2
Reviewer 2 Report
Comments and Suggestions for Authors
Response to the authors:
The authors structured the manuscript presentation in a coherent manner, which now allows for a clear understanding of the work's objective and conclusions. The revised sentences reflect the importance and relevance of their study.